# GPRC6A as a novel kokumi receptor responsible for enhanced taste preferences by ornithine

**Takashi Yamamoto[1]\*, Kayoko Ueji[1], Haruno Mizuta[1], Chizuko Inui-Yamamoto[2], Natsuko Kumamoto[3], Yasuhiro Shibata[3], Shinya Ugawa[3]**

[1]Department of Nutrition, Faculty of Human Sciences, Kio University, Nara, Japan; [2]Department of Oral Anatomy and Developmental Biology, Osaka University Graduate School of Dentistry, Osaka, Japan; [3]Department of Anatomy and Neuroscience, Graduate School of Medical Sciences, Nagoya City University, Nagoya, Japan

## eLife Assessment

In this **valuable** study, the authors used rats to determine the receptor for a food-related perception that has been characterized in humans. The data are **solid** in terms of methods and analysis: the data show that this stimulus (ornithine) has some additive effects in terms of increasing preference and taste response in rats when it is mixed with other more common taste stimuli. Therefore, the combinations of experiments generally support (but do not conclusively prove) the hypothesis that the 'kokumi' taste effect elicited by this stimulus in humans may be mediated by the specific receptor examined in the study.

**\*For correspondence:**
ta.yamamoto@kio.ac.jp

**Competing interest:** The authors declare that no competing interests exist.

**Abstract** The concept of 'kokumi', which refers to an enhanced and more delicious flavor of food, has recently generated considerable interest in food science. However, kokumi has not been well studied in gustatory physiology, and the underlying neuroscientific mechanisms remain largely unexplored. Our previous research demonstrated that ornithine (L-ornithine), which is abundant in shijimi clams, enhanced taste preferences in mice. The present study aimed to build on these findings and investigate the mechanisms responsible for kokumi in rats. In two-bottle preference tests, the addition of ornithine, at a low concentration that did not increase the favorability of this substance alone, enhanced the animals' preferences for umami, sweet, fatty, salty, and bitter solutions, with the intake of monosodium glutamate showing the most significant increase. Additionally, a mixture of umami and ornithine synergistically induced significant responses in the chorda tympani nerve, which transmits taste information to the brain from the anterior part of the tongue. The observed preference enhancement and increase in taste-nerve response were abolished by antagonists of the G-protein-coupled receptor family C group 6 subtype A (GPRC6A). Furthermore, immunohistochemical analysis indicated that GPRC6A was expressed in a subset of type II taste cells in rat fungiform papillae. These results provide new insights into flavor-enhancement mechanisms, confirming that ornithine is a kokumi substance and GPRC6A is a novel kokumi receptor.

## Introduction

Enjoying delicious food is a fundamental pleasure in daily life and contributes toward both physical and mental well-being. The perception of deliciousness requires the harmonious integration of various sensory experiences, including the five basic tastes (sweet, salty, sour, bitter, and umami), the taste

of fat, spices, aroma, texture, temperature, sight, and sound. These inputs are transmitted to the brain via sensory nerves, where sensory modalities and qualitative information are analyzed (*Rolls, 1989*; *Spector and Travers, 2005*). Beyond these cognitive aspects, the brain further processes these sensations to evoke emotional responses, contributing to palatability or unpleasantness (*Small, 2012*; *Wang et al., 2018*; *Yamamoto, 2008*). Therefore, sensory information from substances entering the oral and nasal cavities influences both cognition and emotions.

Conversely, despite having minimal direct taste effects and a lack of clear information for the brain to process, some substances and their receptors can modify taste information mainly at the peripheral level, thereby controlling palatability. Specifically, recent research on taste modification has focused on 'kokumi' substances and receptors (*Ahmad and Dalziel, 2020*; *Kuroda, 2024*; *Yamamoto and Inui-Yamamoto, 2023*). Kokumi is a Japanese word that denotes enhanced tastiness of food comprising complex ingredients. According to *Ohsu et al., 2010*, when kokumi substances such as glutathione or γ-Glu-Val-Gly are added to chicken consommé at low concentrations that do not produce a specific taste of their own, they bind to the calcium-sensing receptor (CaSR) to induce kokumi by enhancing the 'thickness', 'mouthfulness', and 'continuity' of the dish.

These three attributes were originally introduced by *Ueda et al., 1997*, who translated Japanese terms describing the profound characteristics of kokumi, which are deeply rooted in Japanese culinary culture (*Yamamoto and Inui-Yamamoto, 2023*; *Nishimura, 2019*; *Yamamoto, 2019*). However, these simply translated terms have caused global misunderstanding and confusion because they sound like somatosensory rather than gustatory descriptions. Therefore, to clarify that kokumi attributes are inherently gustatory, in the present study we use the terms 'intensity of whole complex tastes (rich flavor with complex tastes)' instead of thickness, 'mouthfulness (spread of taste and flavor throughout the oral cavity)', and 'persistence of taste (lingering flavor)' instead of continuity.

Several studies have been conducted on CaSR agonists as candidate kokumi substances (*Chang et al., 2022*; *Feng et al., 2019*; *Kuroda et al., 2012a*; *Kuroda et al., 2012b*; *Liu et al., 2015*; *Li et al., 2022*; *Salger et al., 2019*; *Shibata et al., 2017*; *Toelstede et al., 2009*), although neuroscientific research on the mechanism of kokumi induction remains limited and underdeveloped (*Bigiani and Rhyu, 2023*; *Rhyu et al., 2020*; *Yamamoto and Mizuta, 2022*; *Maruyama, 2024*; *Yamamoto and Mizuta, 2022*). In contrast, our previous study in mice (*Mizuta et al., 2021*) focused on ornithine, a non-protein amino acid, as a potential kokumi substance and suggested that the G-protein-coupled receptor family C group 6 subtype A (GPRC6A) is responsible for its action, given that ornithine has a high affinity for this receptor (*Wellendorph and Bräuner-Osborne, 2004*; *Christiansen et al., 2007*;

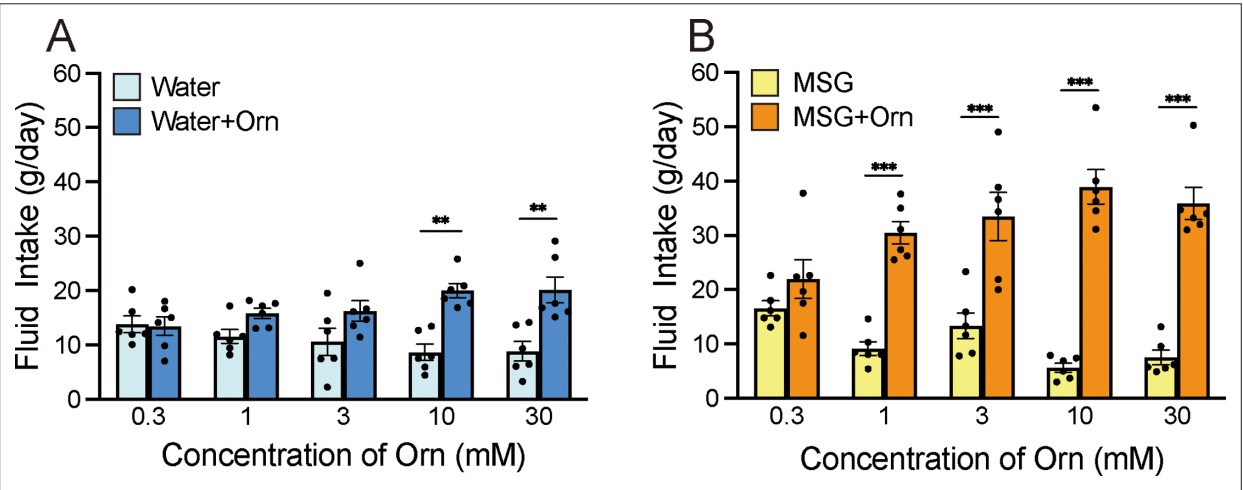

**Figure 1.** Additive effects of ornithine (Orn) at different concentrations. (**A**) Water intake with and without five Orn concentrations. (**B**) Intake of 0.03 M MSG with and without Orn. Each value represents the mean ± SEM; n=6. **$p<0.01$, ***$p<0.001$ (Bonferroni correction). MSG, monosodium glutamate; SEM, standard error of the mean.

The online version of this article includes the following source data for figure 1:

**Source data 1.** Two-bottle preference test.

*Oya et al., 2013*). As this is the only published study to propose GPRC6A as a potential kokumi receptor other than CaSR, this hypothesis needs to be confirmed.

Neuroscientific studies (*Maruyama et al., 2012*; *Yamamoto and Mizuta, 2022*; *Yamamoto et al., 2009*) in mice and rats provide evidence that glutathione and y-Glu-Val-Gly activate CaSRs, and modify behavioral responses to other tastants in a way that may correspond to kokumi as experienced by humans. However, to our best knowledge, only our prior study in mice has examined ornithine as a kokumi candidate (*Mizuta et al., 2021*). Therefore, the current study aimed to investigate the effects of ornithine on sweet, salty, sour, bitter, umami, and fatty taste enhancement in rats and to clarify the involvement of the GPRC6A receptor. Whether rodents are suitable models for kokumi research is unclear; however, if the expression of kokumi flavor is fundamental to the enjoyment of delicious food, there should be basic mechanisms common to humans and rodents, even if a similar human perception of kokumi is difficult to detect in rats. Elucidating these neuroscientific mechanisms may shed light on the elusive nature of kokumi in humans.

## Results

### Two-bottle preference tests in rats

#### Additive effects of ornithine at different concentrations

To investigate whether ornithine has a favorable taste, we conducted a preference test by comparing distilled water (DW) with aqueous solutions containing various ornithine concentrations (*Figure 1A*). Two-way analysis of variance (ANOVA; solution × concentration) revealed a significant main effect of solution [$F_{(1, 50)}=33.93$, $p<0.0001$], no main effect of concentration [$F_{(4, 50)}=0.145$, $P>0.05$], and a significant solution-concentration interaction [$F_{(4, 50)}=4.070$, $p<0.01$]. Post hoc Bonferroni tests showed that the intake of 10 and 30 mM ornithine was significantly greater ($p<0.01$) than that of plain DW, indicating a preference for higher concentrations of ornithine. Next, we examined the additive effect of ornithine at the same concentrations on the preference for 0.03 M monosodium glutamate (MSG; *Figure 1B*). Two-way ANOVA revealed a significant main effect of solution [$F_{(1, 50)}=175.41$, $p<0.0001$] and a solution-concentration interaction [$F_{(4, 50)}=8.371$, $p<0.0001$], with no main effect of concentration [$F_{(4, 50)}=0.899$, $p>0.05$]. Subsequent Bonferroni analyses indicated that the addition of ornithine at concentrations ranging from 1 to 30 mM resulted in significantly greater intake ($p<0.001$) than that for MSG alone. Based on these results, we used 1 mM ornithine to examine its additive effects in subsequent experiments.

#### Additive effects of 1 mM ornithine on intake of different tastants

We examined the intake of eight taste solutions at four concentrations, both alone and in combination with 1 mM ornithine (*Figure 2*). The eight solutions were sweet (sucrose), salty (NaCl), sour (citric acid), bitter (quinine hydrochloride [QHCl]) umami (MSG, monopotassium glutamate [MPG], or inosine monophosphate [IMP]), and fatty (Intralipos). Bonferroni's multiple-comparison analysis revealed that MSG and MPG at all four concentrations were significantly preferred when combined with ornithine ($p<0.05$, 0.01, or 0.001; *Figure 2B and C*). This was also observed for IMP (*Figure 2A*), Intralipos (*Figure 2D*), and sucrose (*Figure 2E*) at two concentrations, as well as NaCl (*Figure 2F*) at one concentration. However, the rats showed no difference in preference for citric acid regardless of the addition of 1 mM ornithine. In contrast, QHCl with ornithine was preferred at three of the four concentrations, suggesting that the aversive taste of QHCl was attenuated by ornithine. Notably, the effects of ornithine differed among the umami solutions; ornithine more effectively increased the preference for MSG and MPG than for IMP.

#### Additive effects of ornithine in brief-exposure tests with and without antagonists

To determine whether the enhancement of preference induced by ornithine was caused by an intra-oral event rather than by post-oral consequences, a brief-exposure (or short-term, 10 minute) two-bottle preference test was conducted. We confirmed that the intake of DW with 1 mM ornithine did not differ from that of plain DW ($p>0.05$; *Figure 3A*). However, the intake of 0.03 M MSG was significantly increased ($p<0.01$) by the addition of 1 mM ornithine (*Figure 3B*). This preference was

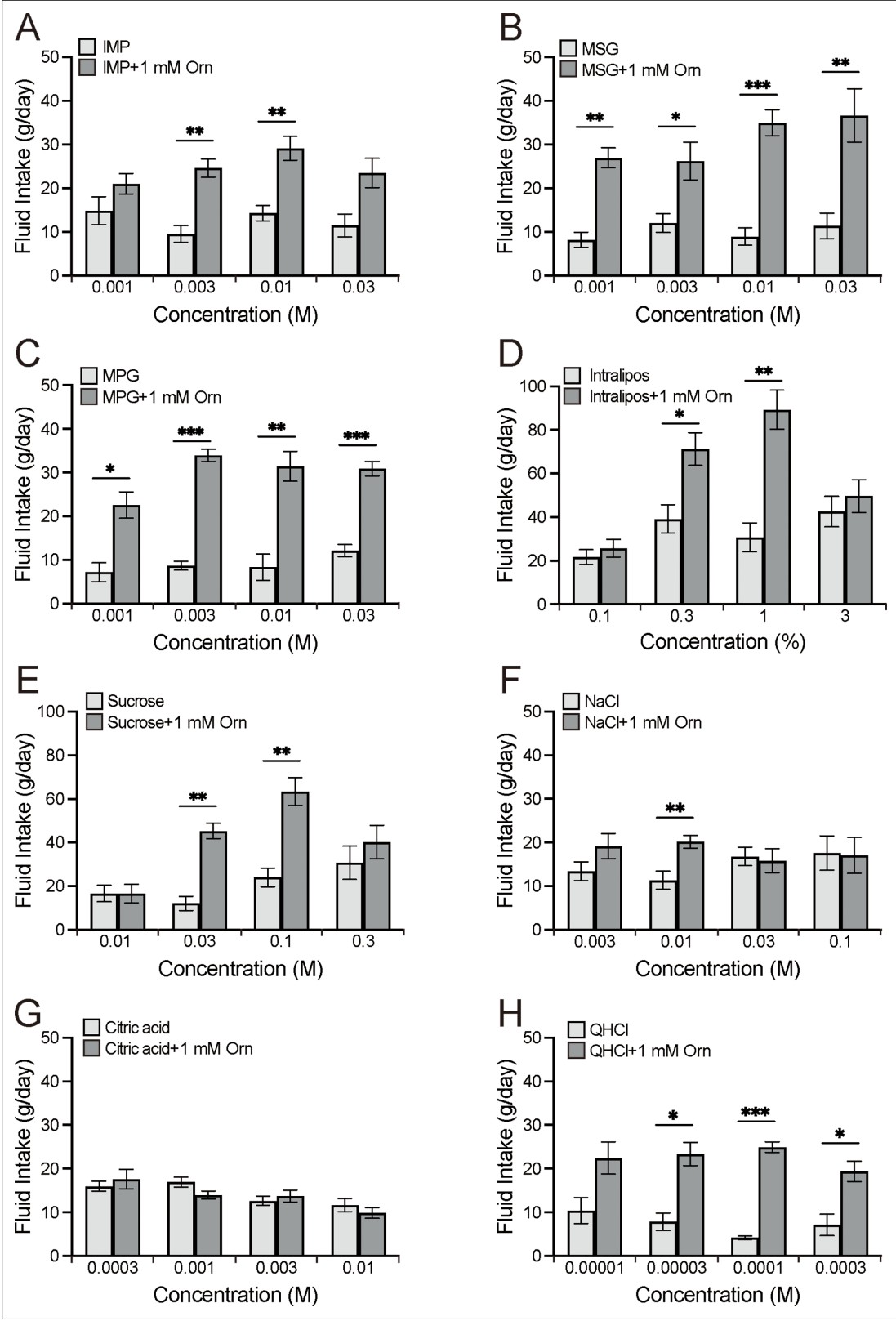

**Figure 2.** Additive effects of 1 mM ornithine (Orn) on fluid intake for eight different taste solutions. Fluid intake with and without Orn is shown for IMP (**A**), MSG (**B**), MPG (**C**), Intralipos (**D**), sucrose (**E**), NaCl (**F**), citric acid (**G**), and QHCl (**H**). Each value represents the mean ± SEM; n=8. *p<0.05, **p<0.01, ***p<0.001 (Bonferroni correction). IMP, inosine monophosphate; MSG, monosodium glutamate; MPG, monopotassium phosphate; QHCl, quinine hydrochloride; SEM, standard error of the mean.

*Figure 2 continued on next page*

*Figure 2 continued*

The online version of this article includes the following source data and figure supplement(s) for figure 2:

**Source data 1.** Two-bottle preference test.

**Figure supplement 1.** Effects of ornithine (L-ornithine, Orn) supplementation of low-sodium (0.7%) miso soup on three kokumi attributes in humans: intensity, mouthfulness, and persistence of taste.

**Figure supplement 1—source data 1.** Human sensory test.

maintained in the presence of 60 µM calindol, a GPRC6A antagonist (*Figure 3C*), but disappeared at a calindol concentration of 300 µM (*Figure 3D*). Similarly, the GPRC6A antagonist epigallocatechin gallate (EGCG) had no effect at a concentration of 30 µM (*Figure 3E*) but abolished the ornithine-enhanced preference at 100 µM (*Figure 3F*).

## Additive effects of ornithine after chorda tympani transection

To determine how taste information from the anterior part of the tongue influences ornithine-induced taste preferences, we compared intake during the long-term two-bottle preference test before and after transection of the chorda tympani (CT). As shown in *Figure 4A and C*, the combination of 1 mM ornithine and 0.03 M MSG was significantly more ingested than MSG alone (p<0.01, paired *t*-test, two-tailed) before both sham and experimental treatments. The same preference was observed following the sham-control operation (*Figure 4B*). However, in the experimental group, CT transection abolished the increased favorability induced by ornithine supplementation (*Figure 4D*).

## Taste-nerve recording in rats

The aqueous solutions of ornithine used in the behavioral experiments induced small CT responses, which increased in a dose-dependent manner, as shown by sample recordings (*Figure 5A*) and a graphical representation of mean response magnitudes (*Figure 5D*). One-way ANOVA revealed significant differences among the responses at different ornithine concentrations [F(4, 15)=14.660, p<0.001]. Additionally, the response to 30 mM ornithine survived the Bonferroni correction for multiple testing, which indicated that this response was significantly higher (p<0.01) than those at lower concentrations. The response to 1 mM ornithine, which was used in the behavioral experiments, was negligible compared to the standard response to NH$_4$Cl. However, when 1 mM ornithine was added to 0.03 M MSG, the response increased significantly (p<0.05) compared with that for plain MSG (*Figure 5E*). A similar additive effect of ornithine was observed when the solution was prepared with 0.1 mM amiloride, a sodium-channel blocker (*Figure 5B and E*), suggesting that glutamate, rather than sodium ions, was responsible for this increased response. One-way ANOVA revealed significant differences among the relative response values shown in *Figure 5E* [F(3, 16)=9.174, p<0.001]. Finally, we examined the effect of calindol on the increased response to MSG with ornithine. As shown by the sample recordings (*Figure 5C*) and graphical representation (*Figure 5F*), the increased response to MSG with the addition of ornithine (p<0.01) was no longer observed when 300 µM calindol was added. One-way ANOVA revealed significant differences among these responses visualized in *Figure 5F* [F(2, 9)=4.690, p<0.05]. EGCG (100 µM), another GPRC6A antagonist, showed effects similar to those of calindol (data not shown).

## Immunohistochemical localization of GPRC6A in rat taste cells

Immunohistochemical analyses were performed to determine whether GPRC6A was expressed in the taste cells of rat fungiform, foliate, and circumvallate taste buds. In the fungiform papillae, a small number of spindle-shaped taste cells exhibited GPRC6A-immunoreactivity (*Figure 6A*). In the foliate and circumvallate papillae, GPRC6A-immunopositive taste cells were barely detectable, and GPRC6A-expressing cells were likely to constitute less than 1% of the total taste cell population in the respective taste papillae (*Figure 6B and C*). These results demonstrated that GPRC6A was preferentially located in subpopulations of fungiform taste cells in the rat.

Since rat taste cells comprise several cell types, we examined the colocalization of GPRC6A and cell type-specific markers in the fungiform papillae. A small subpopulation of GPRC6A-immunopositive cells was found to be immunoreactive for IP3R3, a marker for the majority of the type II cell population (*Figure 7A*). Approximately 11% of GPRC6A-positive cells overlapped with IP3R3 (9 double-positive

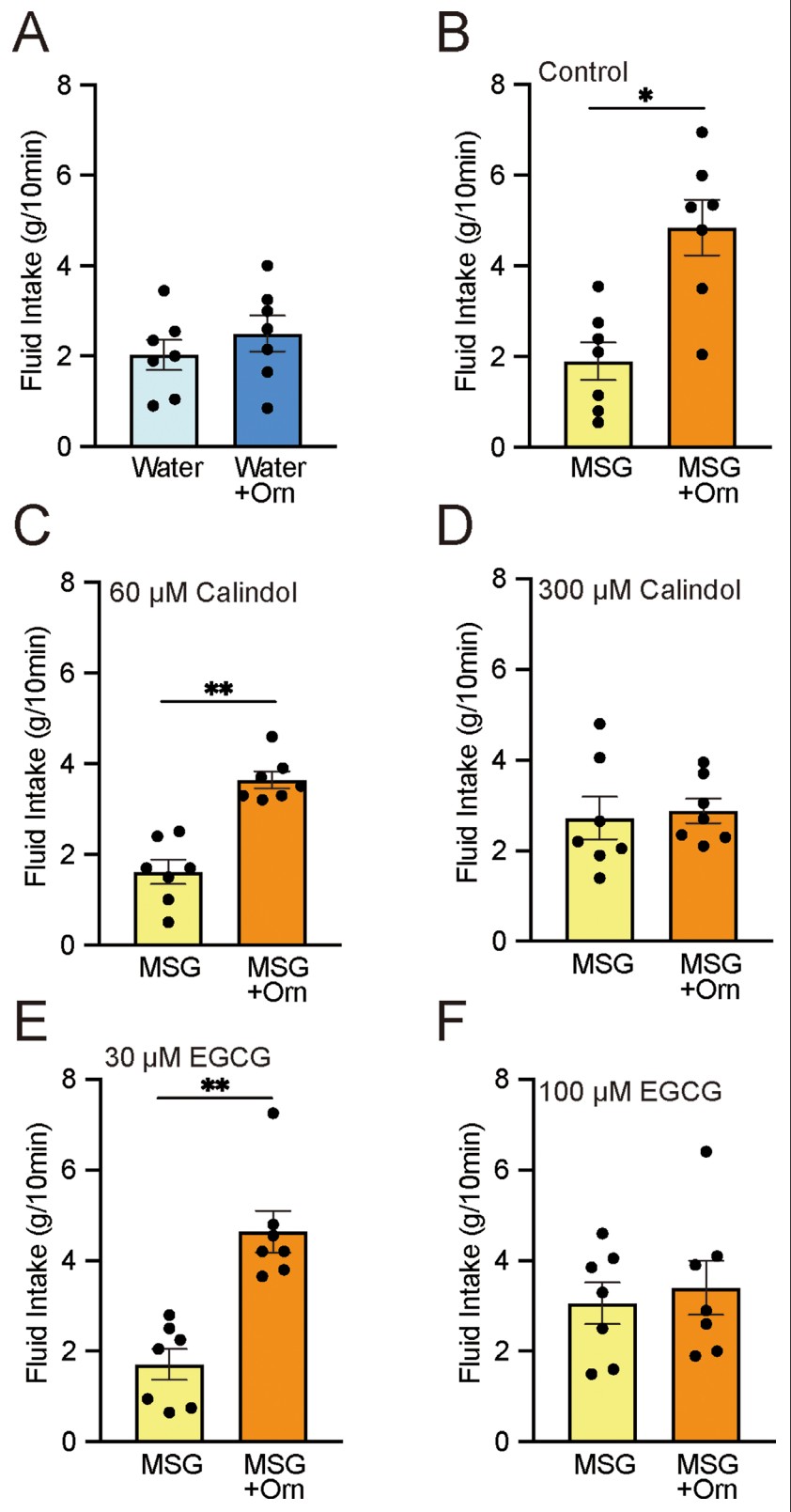

**Figure 3.** Brief-exposure (10 min) two-bottle preference test results for solutions with and without 1 mM ornithine (Orn) and effects of GPRC6A antagonists calindol and EGCG. (**A**) Intake of water with and without Orn. (**B**) Intake of 0.03 M MSG with and without Orn. (**C**) Intake of MSG with and without Orn in 60 μM calindol. (**D**) Intake of MSG with and without Orn in 300 μM calindol. (**E**) Intake of MSG with and without Orn in 30 μM EGCG. (**F**) Intake

*Figure 3 continued on next page*

*Figure 3 continued*

of MSG with and without Orn in 100 µM EGCG. Each value represents the mean ± SEM; n=7. *p<0.05, **p<0.01 (paired *t*-test, two-tailed). MSG, monosodium glutamate; EGCG, epigallocatechin gallate; GPRC6A, G-protein-coupled receptor family C group 6 subtype A; SEM, standard error of the mean.

The online version of this article includes the following source data and figure supplement(s) for figure 3:

**Source data 1.** Brief-exposure two-bottle preference test.

**Figure supplement 1.** Long-term exposure (1 day) two-bottle preference test results for solutions with and without 1 mM ethyl gallate, a GPRC6A agonist, and effects of 100 µM EGCG, a GPRC6A antagonist.

**Figure supplement 1—source data 1.** Two-bottle preference test with and without gallate.

cells/80 GPRC6A-positive cells), while approximately 8.3% of IP3R3-positive cells expressed GPRC6A (9 double-positive/109 IP3R3-positive cells). In contrast, GPRC6A-positive cells were unlikely to colo-calize with a-gustducin, another marker for a subset of type II cells, in single taste cells (0 double-positive cell/93 GPRC6A-positive cells) (*Figure 7B*). Regarding type III cell markers, GPRC6A-positive cells were unlikely to colocalize with 5-hydroxytryptamine (serotonin, 5-HT) in single taste cells (0 double-positive cell/75 GPRC6A-positive cells) (*Figure 7C*). As synaptosomal-associated protein 25 kDa (SNAP-25) labels not only type III cells but also the dense network of intragemmal nerve fibers, which are nerve fibers that extend directly into the structure of the taste bud to transmit taste signals from taste cells to the brain (*Tizzano et al., 2015*), it is difficult to detect SNAP-25-expressing taste cells surrounded by intense SNAP-25-immunoreactivity of the nerve fibers. However, as far as we investigated, no SNAP-25-expressing cells were found among GPRC6A-positive cells examined (0 double-positive cell/104 GPRC6A-positive cells) (*Figure 7D*). These results indicate that GPRC6A is preferentially localized in a subpopulation of IP3R3-expressing taste cells in rat fungiform papillae and suggest that GPRC6A exerts taste-modifying activity in at least some type II cells in the papillae.

## Discussion

In this study, we used a 24-hour two-bottle preference test in rats to demonstrate that the addition of 1 mM ornithine (which itself did not exhibit any taste preference) to tastants comprising umami substances, Intralipos (representing fatty taste), sucrose (sweetness), NaCl (saltiness), and QHCl (bitterness) increased the intake of each solution, suggesting enhanced palatability, especially to umami stimuli. Similar increases in ingestion were observed in a short-term two-bottle preference test using MSG supplemented with ornithine, indicating that this effect was due to the taste sensation in the oral cavity rather than post-ingestive effects. Furthermore, the disappearance of ornithine's effect on MSG following the application of GPRC6A antagonists and transection of the CT nerve, which innervates taste buds in the anterior tongue, suggests that GPRC6A, which was expressed in the taste cells in this area, is involved in this effect. Supporting this finding, the response of the CT nerve to MSG increased with ornithine supplementation, and the action of a GPRC6A antagonist abol-ished this change. In addition, immunohistochemical staining showed that GPRC6A predominantly appeared in the fungiform papillae of the anterior tongue rather than in the foliate or circumvallate papillae of the posterior tongue. Despite the differences described below, these results align with our previous findings in mice (*Mizuta et al., 2021*). Together with those from our human study (*Figure 2—figure supplement 1*), these results confirm that ornithine is a potential kokumi substance and that GPRC6A is a candidate kokumi receptor. These insights provide basic neuroscientific knowledge for interpreting the observation in humans that adding ornithine to miso soup, which has a rich umami flavor with slightly sweet and salty tastes, enhances the three elements of kokumi: intensity of whole complex tastes, mouthfulness (spread of taste in the oral cavity), and persistence of taste.

Several similarities and differences are apparent when comparing the results obtained with those of our previous study in mice (*Mizuta et al., 2021*). In both studies, the same concentration of ornithine was used for supplementation (1 mM). The addition of ornithine increased the favorability of basic tastes including umami, sweet, fatty, salty, and bitter tastes, with a more pronounced preference for MSG than IMP among the umami substances. However, while a weak increase in citric acid preference was observed in mice, no such effect was identified in rats. The effects of GPRC6A antagonists and the CT-nerve responses were consistent between the two species. However, the expression of GPRC6A

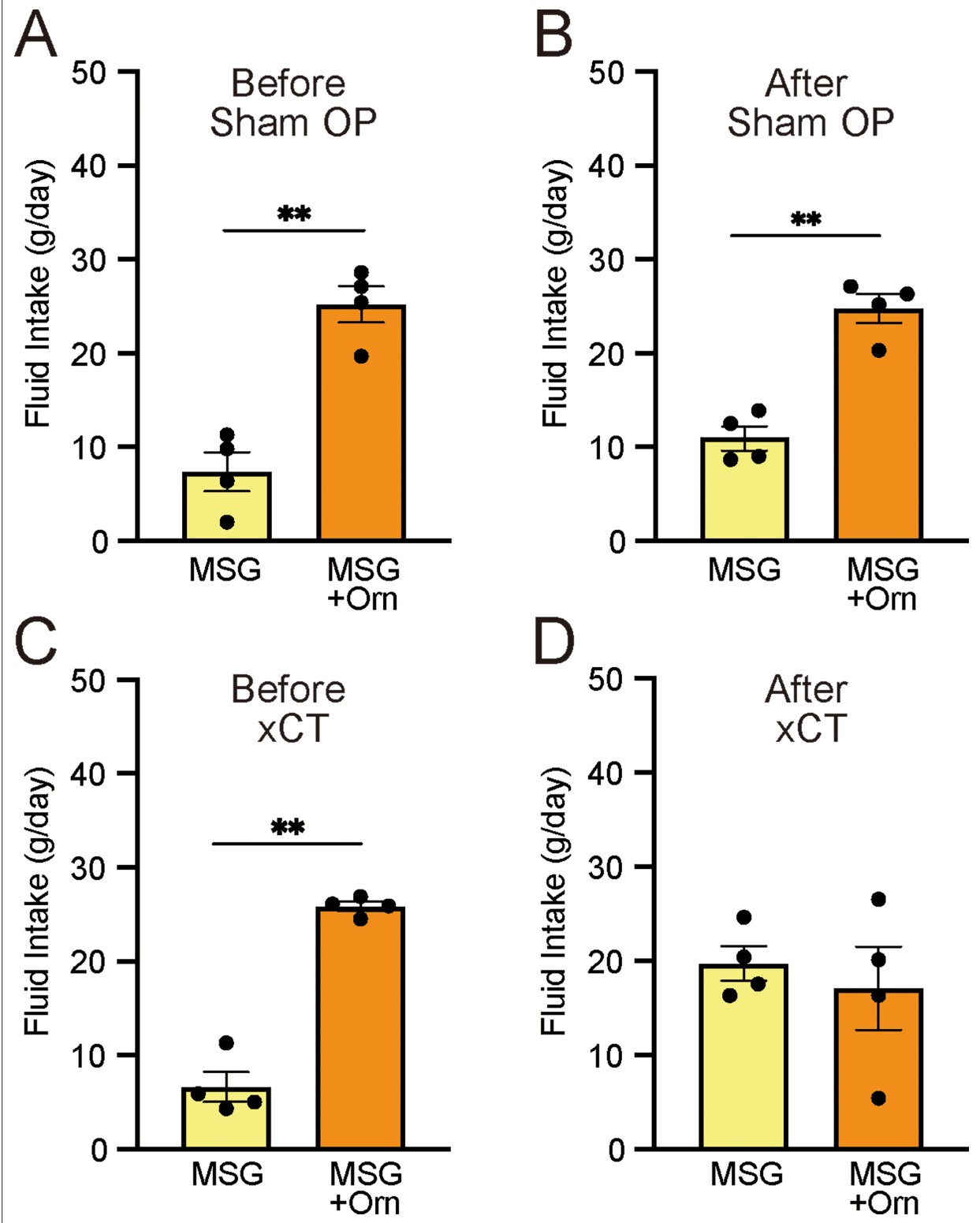

**Figure 4.** Effects of bilateral chorda tympani transection (xCT) and control sham operations (Sham OP) on intake of 0.03 M MSG with and without 1 mM ornithine (Orn). Fluid intake before (**A, C**) and after (**B, D**) the operations. Each value represents the mean ± SEM; n=4. **p<0.01 (paired *t*-test, two-tailed). MSG, monosodium glutamate; SEM, standard error of the mean.

The online version of this article includes the following source data for figure 4:

**Source data 1.** Two-bottle preference test after CT transection.

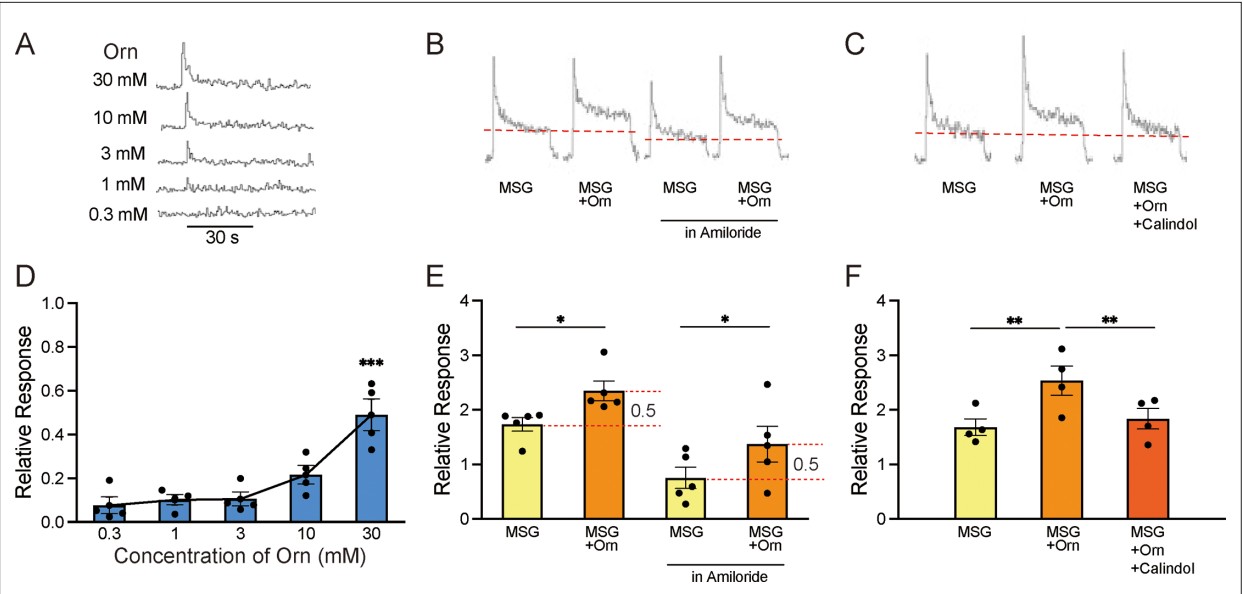

**Figure 5.** Sample recordings of chorda tympani (CT) responses and quantitative representations of mean response magnitudes. (**A**) Nerve responses to five concentrations of ornithine (Orn). (**B**) Nerve responses to 0.03 M MSG with and without 1 mM Orn in water and in 0.01 mM amiloride (sodium-channel blocker). (**C**) Nerve responses to MSG, MSG with Orn, and MSG with Orn in 300 μM calindol. (**D–F**) Graphical representations of the mean magnitudes of CT responses corresponding to (**A**), (**B**), and (**C**), respectively. Each value represents the mean ± SEM normalized to the response to 0.1 M $NH_4Cl$ = 1.0; n=4 or 5. Note the different scaling of the ordinate in (**D**) and (**E**). *$p<0.05$, **$p<0.01$, ***$p<0.001$ (**D** and **F**, Bonferroni correction; **E**, two-tailed paired $t$-test). MSG, monosodium glutamate; SEM, standard error of the mean.

The online version of this article includes the following source data for figure 5:

**Source data 1.** Taste nerve responses.

differed considerably; in mice, this protein was expressed throughout the tongue, particularly in the posterior part, whereas in rats, it was predominantly localized to the anterior region of the tongue. Finally, another major difference is that while mice avoided high concentrations of ornithine, rats seemed to prefer them.

Regarding the increased preference for the umami substances IMP, MSG, and MPG when supplemented with ornithine, one may naturally interpret this as an ornithine-enhanced umami response. However, the possibility that umami substances enhance the response to ornithine should be considered. Depending on its concentration, ornithine may stimulate at least three receptors, namely GPRC6A, CaSR (*Conigrave et al., 2000*; *Shin et al., 2020*), and the conventional amino acid receptor, the T1R1/T1R3 heterodimer (*Nelson et al., 2002*). At the low concentration of 1 mM used in the present study, GPRC6A may be exclusively stimulated, whereas higher concentrations may also stimulate T1R1/T1R3, thereby transmitting ornithine taste information via this receptor. Compared with GPRC6A, CaSR may have limited to no involvement, since EGCG, a GPRC6A-specific antagonist, significantly abolished the increased preference induced by ornithine. Umami substances may promote the binding of ornithine to these receptors. For example, *Tanase et al., 2022* showed that the taste intensity of ornithine increased in the presence of IMP in humans. Similarly, *Ueda et al., 1997* and *Dunkel et al., 2007* reported that the taste thresholds for glutathione and glutamyl peptides, agonists of CaSR, were reduced by umami solutions. In rats, ornithine has a favorable taste, and its palatability increased with its concentration (see *Figure 1A*). When added to ornithine, MSG may enhance the response to ornithine, thereby increasing its palatability. This concept also explains the mixed effects of ornithine and MSG observed in mice. The taste of ornithine itself was increasingly disliked as its concentration was raised (*Mizuta et al., 2021*). Therefore, although the addition of ornithine at concentrations below 3 mM increased palatability, its favorability ceased at concentrations of 10 and 30 mM. Taste information sent by ornithine as its concentration increases, whether pleasant or unpleasant, is likely mediated by the conventional amino acid receptor T1R1/T1R3 rather than GPRC6A. Thus, umami substances may facilitate ornithine stimulation of both kokumi and amino acid

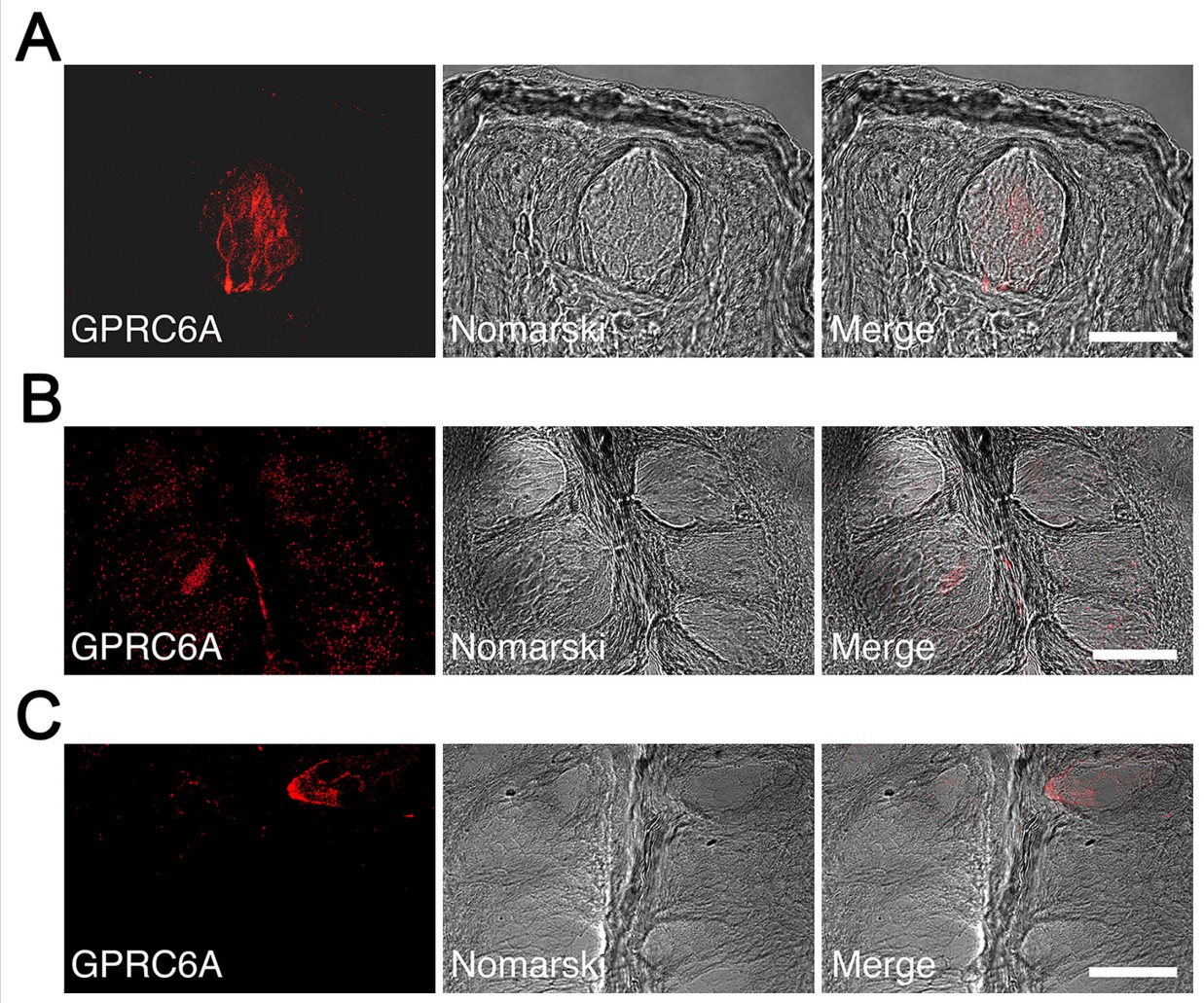

**Figure 6.** Immunohistochemical localization of GPRC6A in rat taste papillae. (**A**) A significant number of spindle-shaped taste cells exhibited intense GPRC6A immunoreactivity in the fungiform papillae. (**B, C**) GPRC6A-immunopositive taste cells were barely detectable in the foliate (**B**) or circumvallate (**C**) papillae. Left panels show GPRC6A in red, middle panels show Nomarski images of the left panels, and right panels show merged images of respective left and middle panels. Scale bars, 50 µm. GPRC6A, G-protein-coupled receptor family C group 6 subtype A.

receptors in a concentration-dependent manner. Ornithine and umami substances interact to produce synergistic effects on palatability (*Yamamoto and Inui-Yamamoto, 2023*).

The essence of kokumi expression is the enrichment of tastes associated with deliciousness (*Yamamoto and Inui-Yamamoto, 2023*). In our experiments with rats (present study) and mice (*Mizuta et al., 2021*), adding ornithine to favorable-tasting solutions such as umami, sweet, and fatty solutions increased their tastiness. This can be explained by the enhanced electrophysiological responses to these tastes induced by ornithine, as evidenced by increased CT-nerve responses for MSG in rats and mice and for other tastes in mice. Conversely, the increased palatability of the aversive bitter taste in both species suggests that ornithine suppresses the response to bitterness. Although the addition of ornithine to quinine slightly decreased the CT-nerve response in mice, this difference was not statistically significant. Reports have shown that ornithine (*Tokuyama et al., 2006*; *Ogawa et al., 2005*) and arginine (*Pi et al., 2018*), which is structurally highly similar to ornithine, both decrease the bitter taste sensation in humans; however, the underlying mechanism is not well understood. We conducted additional experiments in rats using gallate (*Figure 3—figure supplement 1*), which has been reported as an agonist of GPRC6A (*Pi et al., 2018*; *Melis and Tomassini Barbarossa, 2017*). Similar to ornithine, gallate increased the palatability of MSG and quinine, although this effect was abolished by EGCG. Notably, this suggests that GPRC6A agonists may augment responses to favorable tastes such as

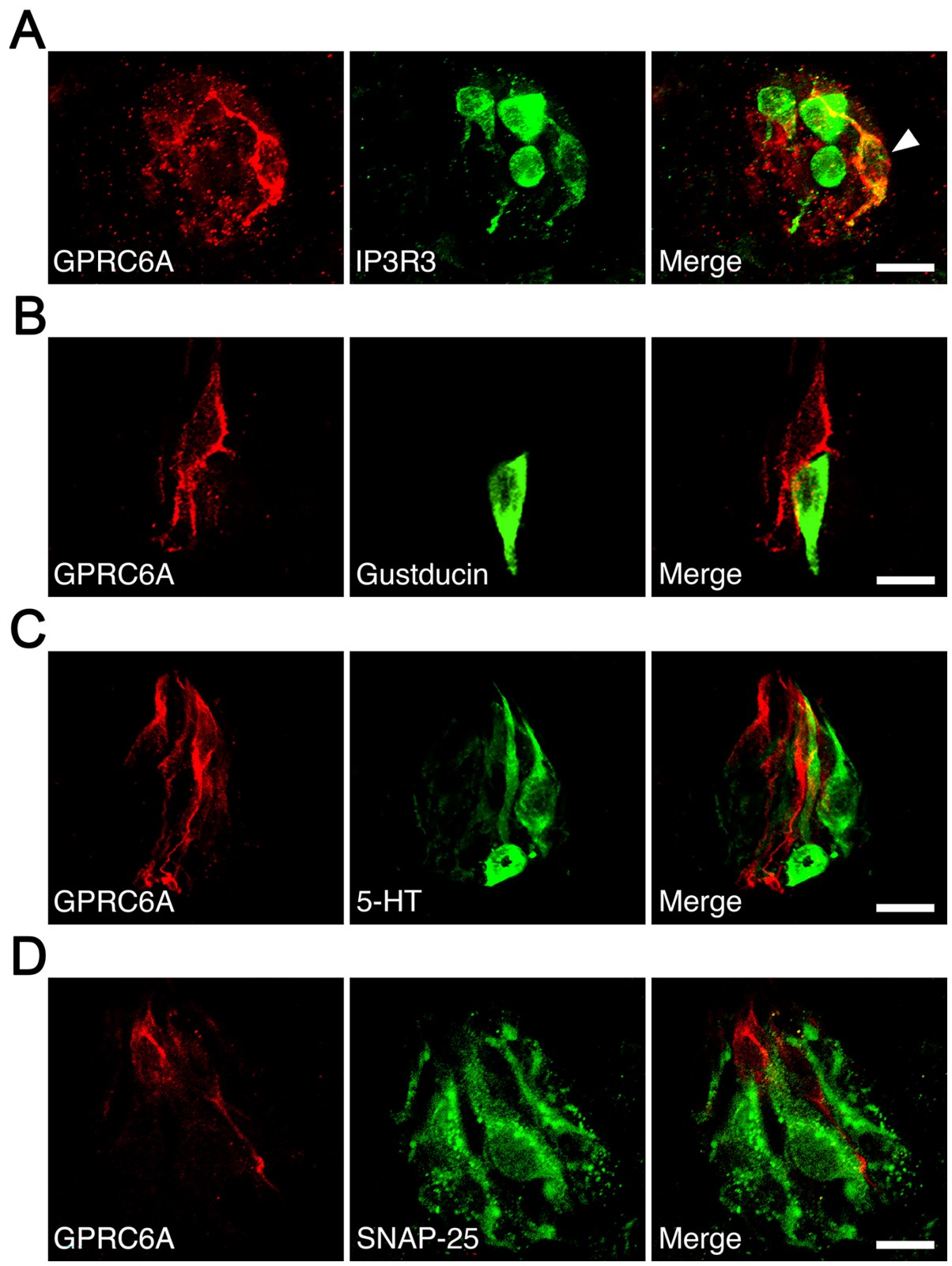

**Figure 7.** Immunohistochemical analysis of colocalization of GPRC6A and cell type-specific markers in taste cells of rat fungiform papilla taste buds. (**A**) Some but not all GPRC6A-immunopositive cells exhibited immunoreactivity for IP3R3, a marker of most type II cells. White arrow indicates GPRC6A/ IP3R3 double-positive taste cell. (**B**) α-Gustducin, another marker of a subset of type II cells, was unlikely to colocalize with GPRC6A in single taste cells. (**C, D**) Neither 5-HT (**C**) nor SNAP25 (**D**), specific markers of type III taste cells, were likely to colocalize with GPRC6A in single taste cells. Scale bars,

*Figure 7 continued on next page*

*Figure 7 continued*

10 µm. GPRC6A, G-protein-coupled receptor family C group 6 subtype A; IP3R3, inositol 1,4,5-trisphosphate receptor type 3; 5-HT, 5-hydroxytryptamine; SNAP25, synaptosomal-associated protein 25 kDa.

The online version of this article includes the following figure supplement(s) for figure 7:

**Figure supplement 1.** Immunohistochemical colocalization of GPRC6A and CaSR in rat fungiform taste papillae.

umami and sweetness, while inhibiting responses to bitter tastes, leading to an overall enhancement of deliciousness.

The CaSR agonist γ-Glu-Val-Gly similarly enhanced the preference for umami, sweet, and fatty tastes in rats, but showed no effect on the favorability of salty, sour, or bitter tastes (*Yamamoto and Mizuta, 2022*). Additionally, while ornithine increased the preference for MSG more than that for IMP, γ-Glu-Val-Gly had the opposite effect. Unlike those of ornithine, the effects of γ-Glu-Val-Gly were not altered by severance of the CT nerve, likely because CaSR is expressed in both the anterior and posterior parts of the tongue (*San Gabriel et al., 2009*). The results of the present study, along with those of our previous animal experiments, clearly demonstrate that the binding of kokumi substances to their receptors triggers the expression of kokumi. However, the mechanism by which this binding enhances the response to umami, fatty, sweet, and other tastes remains unknown. Considering our observation that GPRC6A was co-expressed in type II taste cells positive for IP3R3, some of these cells may be affected by GPRC6A activation. However, it is difficult to explain why co-expression with α-gustducin, which binds to taste receptors in type II cells, was not observed. While we did not examine the co-expression of GPRC6A with the T1R or T2R families, *Maruyama et al., 2012* reported that CaSR is not co-expressed with receptors for umami, sweet, or other tastes in mice. Some form of intercellular communication within taste buds that affects these tastes is possible, although the detailed mechanism requires further research (*Maruyama, 2024*). Whether GPRC6A and CaSR colocalize to the same taste cells should also be considered; however, we have not yet identified any taste cells in which these proteins are co-expressed in our ongoing experiments (*Figure 7—figure supplement 1*). In addition to the above, it is noteworthy that kokumi-active γ-glutamyl peptides have a stronger affinity for the T1R1-MSG receptor complex than for the T1R2-sucrose receptor complex, as demonstrated by molecular modeling approaches (*Yang et al., 2022*). This suggests that these peptides exhibit a greater umami-enhancing effect without the involvement of kokumi-specific receptors.

While many candidate kokumi substances have been reported (*Kuroda, 2024*; *Chang et al., 2022*; *Feng et al., 2019*; *Liu et al., 2015*; *Li et al., 2022*; *Salger et al., 2019*; *Shibata et al., 2017*; *Toelstede et al., 2009*; *Bigiani and Rhyu, 2023*), we are currently exploring the underlying neuroscientific mechanisms using CaSR agonists such as glutathione and γ-Glu-Val-Gly, and GPRC6A agonists such as ornithine and gallate. A common finding in our animal experiments is that these substances enhanced umami to a greater extent than other tastes. They also enhanced sweet and fatty tastes, and to a lesser extent, saltiness. CT recordings indicate that the effects of kokumi stimuli on taste enhancement occur primarily in the peripheral taste organs. When taste responses are individually enhanced and the resulting sensory signals are then transmitted to the brain, where they are processed by the central gustatory and flavor systems, the 'intensity of whole complex tastes (or rich flavor with complex tastes)' associated with kokumi may arise. Kokumi substances may reduce the threshold concentrations as well as increase the suprathreshold responses of tastants. Once the threshold concentrations are lowered, additional taste cells in the oral cavity become activated, and this information is transmitted to the brain. As a result, the brain perceives this input as coming from a wider area of the mouth, leading to the sensation of 'mouthfulness'. Regarding the 'persistence of taste (lingering flavor)', it is common sensory physiological knowledge that a strong stimulus causes a large response that takes more time to return to baseline, thereby inducing a longer-lasting sensation (e.g., *Kawasaki et al., 2016*). Using a kokumi substance to enhance umami, a taste known for its mouthfulness and lingering quality (*Yamamoto and Inui-Yamamoto, 2023*; *Kawasaki et al., 2016*; *Yamaguchi, 1998*), further intensifies these characteristics. The comparatively prolonged aftertaste of umami is due to the dominance of umami receptors in taste buds located in the grooves of the circumvallate and foliate papillae at the back of the tongue, where umami substances are less likely to be washed away. This is evidenced by stronger and more specific umami responses from the posterior rather than the anterior part of the tongue in mice (*Danilova and Hellekant, 2003*; *Ninomiya and Funakoshi, 1989*) and

primates (*Yamaguchi, 1998*; *Hellekant et al., 1997*). Moreover, MSG binds deeply within the Venus flytrap domain of the umami receptor (*Zhang et al., 2008*), making it difficult to dislodge.

In conclusion, when kokumi substances are present in complex foods containing umami substances, their interaction primarily enriches the umami taste, followed by sweet and fatty tastes, and to a lesser extent salty and possibly other tastes. This enhancement is at the core of richness, which is accompanied by the secondary sensations of mouthfulness and persistence of taste. Since ornithine is abundant in foods such as shijimi clams (*Corbicula japonica*) (*Uchisawa et al., 2004*; *Wu and Shiau, 2002*), cheese (*Renes et al., 2019*), and shimeji and maitake mushrooms (*Oka et al., 1984*; *Wagemaker et al., 2007*), we can enjoy the kokumi-enhanced flavor of cuisines that incorporate these ingredients.

## Materials and methods

### Animals

Eight-week-old male Wistar rats were obtained from Japan SLC (Shizuoka, Japan). The rats were individually housed in home cages within a temperature- (25°C) and humidity-controlled (60%) room. A 12:12 hour light/dark cycle was followed, with lights on at 6:00 am and experiments conducted during the light cycle. The animals had free access to food (CE-2 Rodent Diet; CLEA Japan, Inc, Tokyo, Japan) and tap water, except during the brief tests described below in which food access was partially restricted. All animal care and experimental procedures were performed in accordance with the National Institutes of Health guidelines. Experimental protocols were approved by the Institutional Animal Care and Use Committee of Kio University (protocol no. H28-01).

### Behavioral experiment: Two-bottle preference tests

Each rat was trained to drink DW from a stainless-steel spout connected to a plastic bottle. After a 1-week training period, short- and long-term two-bottle preference tests were conducted, in which two bottles were simultaneously presented to each cage. Each stainless-steel spout, designed to minimize dripping with an internal ball, had an inner diameter of 6 mm and was separated from the center of each spout by 5 cm. Each bottle contained the same taste stimulus (or DW), although one of the two was mixed with ornithine (L-ornithine; Kanto Chemical, Tokyo, Japan).

In the long-term test, the two bottles were switched after 24 hours of the 48-hour test session to account for any potential positional preference. In the short-term (or brief exposure) test, the two bottles with spouts 1 cm apart were presented to the animals for 10 minutes after an overnight water deprivation. Thereafter, the animals had free access to water until 6:00 pm, after which the water was removed overnight. The next day, the bottles were switched, and the 10-minute test session was repeated. The bottles containing the fluids were weighed before and after testing to measure the intake volume. The total intake volume over 48 hours or 20 minutes was divided by 2 to obtain the intake volume per day or 10 minutes for the long- and short-term tests, respectively.

Six basic taste stimuli were used: sucrose, NaCl, citric acid, QHCl, MSG (all Kanto Chemical), and Intralipos (a parenteral-stable soybean oil emulsion; Otsuka Pharmaceutical Factory, Tokyo, Japan). In addition to MSG, MPG and IMP (both Ajinomoto, Tokyo, Japan) were also used as umami substances. The taste stimuli were dissolved in DW at various concentrations. A range of concentrations of individual tastants was presented to the same group of animals, starting with the lowest concentration. Ornithine was typically added at a concentration of 1 mM, with other concentrations used as required. In some experiments, ethylene gallate (Kanto Chemical) was used instead of ornithine. A sodium-channel blocker, amiloride (Sigma-Aldrich, Tokyo, Japan), was used to reduce the sodium response to MSG.

For GPRC6A antagonists, we used NPS-2143 (ChemScene, Monmouth Junction, NJ, USA), calindol (Cayman Chemical, Ann Arbor, MI, USA), and EGCG (Tokyo Chemical Industry, Tokyo, Japan). NPS-2143 and calindol were dissolved in 99.5% ethanol at a concentration of 0.1% and diluted to various concentrations using DW. EGCG was prepared in DW.

A total of 97 rats was divided into two groups of 6, three groups of 7, and eight groups of 8 for the taste-preference tests. The first two groups were used for long-term exposure tests of (1) DW vs. DW with different ornithine concentrations, and (2) 0.03 M MSG vs. 0.03 M MSG with different ornithine concentrations, respectively. The next three groups were used for respective brief-exposure tests of (1) DW vs. DW with 1 mM ornithine and 0.03 M MSG vs. 0.03 M MSG with 1 mM ornithine, (2) MSG

vs. MSG with ornithine, both containing 60 or 300 μM calindol, and (3) MSG vs. MSG with ornithine, both containing 30 or 100 μM EGCG. The final eight groups underwent long-term tests. Six of these were used to assess different concentrations of IMP, MSG, MPG, sucrose, Intralipos, and NaCl with and without 1 mM ornithine, respectively. One group was used to test both citric acid and QHCl with and without 1 mM ornithine. The last group was used to test both MSG and QHCl with and without 1 mM ethylene gallate.

When examining the range of ornithine and tastant concentrations, we presented the solutions to the animals starting with the lowest concentration. If two different tastants were tested in one group, as in the first of the brief-exposure tests and the long-term test for citric acid and QHCl, the order of presentation was counterbalanced. Statistical analyses were performed within groups or between taste stimuli with and without ornithine, but not across different groups or taste stimuli.

## Transection of the CT nerve

Eight naïve rats were randomly divided into transection and sham-operation groups (four rats each). They underwent the long-term two-bottle preference test as described above, with the bottles containing 0.03 M MSG with and without 1 mM ornithine. Thereafter, the animals were anesthetized via intraperitoneal injection of a combination anesthetic (0.3 mg/kg medetomidine, 4.0 mg/kg midazolam, and 5.0 mg/kg butorphanol). In the transection group, the ear ossicles through which the CT innervates the taste buds on the anterior part of the tongue were removed bilaterally, whereas in the sham-operation group, the operation was stopped immediately before the removal of ear ossicles. Postoperatively, the rats were injected with penicillin G sodium (100 mg/kg) to prevent infection. After 6 days of recovery, the same rats were subjected to the long-term two-bottle preference test using the same taste stimuli. The mean group preferences were then compared. After the experiment, transection was confirmed by microscopic verification of the loss of taste buds on the tongue.

## Taste-nerve recordings

Five naïve rats were deeply anesthetized as described above. Each rat was tracheotomized and secured in a head-holder. The left CT nerve was exposed using a lateral approach (*Yamamoto and Kawamura, 1972*), excised as it exited the tympanic bulla, and dissected from the underlying tissue. The nerve was placed onto a platinum-wire recording electrode (0.1 mm diameter), while an indifferent electrode was placed in contact with a nearby exposed tissue. The responses were processed using a bandpass filter with cutoff frequencies ranging from 40 Hz to 3 kHz and visualized using an oscilloscope (VC11; Nihon Kohden, Tokyo, Japan). The responses were fed to a digitally controlled summator (*Walsh and Halpern, 1974*). The number of discharges was summed over 500 ms epochs using a spike counter (DSE-345; DIA Medical System, Tokyo, Japan) to derive summated responses. The data were stored on a PC, and the total spikes over the entire 30-second stimulus period were counted using the PowerLab system (PowerLab/4SP; ADInstruments, Bella Vista, NSW, Australia) for quantitative analyses.

Each taste stimulus (3 mL) was applied to the anterior dorsal tongue for 30 seconds, followed by rinsing with DW for at least 60 seconds. The response to each stimulus was expressed relative to the magnitude of responses to 0.1 M $NH_4Cl$. The analyses of CT results were based on the average values from at least three repeated trials in individual animals.

## Immunohistochemistry

Male Wistar rats (8–12 weeks old) were deeply anesthetized with isoflurane and transcardially perfused with saline followed by 2% paraformaldehyde in 0.1 M phosphate buffer. To label serotonin-accumulating type III taste cells (*Yee et al., 2001*), rats were injected with 5-hydroxy-l-tryptophan at 80 mg/kg body weight 1 hour before anesthesia. Their tongues were dissected out, soaked in 20% sucrose/0.01 M phosphate-buffered saline (PBS) overnight at 4°C, embedded in optimal-cutting-temperature compound (Sakura Finetek, Tokyo, Japan), and cut into 20 μm thick sections using a cryostat. The sections were immersed in 0.01 M PBS containing 0.2% Triton X-100 and incubated overnight at 4°C with rabbit anti-GPRC6A antibody (1:50; orb385435; Biorbyt, Cambridge, UK) in combination with either goat anti-IP3R3 (1:100; NB100-2545; Novus, Centennial, CO, USA), anti-α-gustducin (1:500; LSB4942; LSBio, Seattle, WA, USA), anti-5-HT (1:100; ab66047; Abcam, Cambridge, UK), anti-SNAP25 antibody (1:100; ab31281; Abcam), or mouse anti-CaSR antibody (1:100; ab19347; Abcam)

diluted in the same Triton X-100-containing solution. This was followed by Alexa Fluor 594-conjugated anti-rabbit immunoglobulin G (IgG; 1:500; A-21207; Thermo Fisher Scientific, Waltham, MA, USA) and 488-conjugated anti-goat IgG secondary antibodies (1:500; A-11055; Thermo Fisher Scientific). After washing, the sections were cover-slipped with Fluoromount (Diagnostic BioSystems, Pleasanton, CA, USA) and imaged using a Nikon A1Rs confocal laser scanning microscope (Nikon, Tokyo, Japan). The specificity of the anti-GPRC6A antibody was verified as previously described (*Mizuta et al., 2021*) (data not shown). IP3R3 is a marker protein in most type II taste cell populations (*Sekerková et al., 2005*). Further, α-gustducin, 5-HT, and SNAP25 are markers of a subset of types II, III, and III taste cells, respectively (*Mizuta et al., 2021*; *Yee et al., 2001*; *Sekerková et al., 2005*).

### Human sensory testing

We recruited 22 participants (19 women and 3 men, aged 21–28 years) from Kio University who were not affiliated with our laboratory, including students and staff members. All participants passed a screening test based on taste sensitivity. According to the responses obtained from a pre-experimental questionnaire, we confirmed that none of the participants had any sensory abnormalities, eating disorders, or mental disorders, or were taking any medications that may potentially affect their sense of taste. All participants were instructed not to eat or drink anything for 1 hour prior to the start of the experiment. We provided them with a detailed explanation of the experimental procedures, including safety measures and personal data protection, without revealing the specific goals of the study. Thereafter, each participant provided written informed consent. This study was approved by the Ethics Committee of Kio University (approval no. H30-10), and all experiments adhered to the principles of the Declaration of Helsinki.

Miso soup with 0.7% salt was prepared by dissolving commercial miso paste (Tokujyo; Takeya Miso Co., Suwa, Japan) in hot tap water (control soup). Subsequently, three test soups were prepared by dissolving 1, 3, or 10 mM L-ornithine in the control soup. An aliquot of 30 mL of each test soup was randomly served in a paper cup along with a cup of control soup. Intensity of taste, mouthfulness, and persistence of taste were evaluated according to *Ohsu et al., 2010*. Intensity was expressed as the increased taste intensity 5 seconds after tasting. Persistence was defined as lingering taste intensity 20 seconds after tasting. Finally, mouthfulness was described as the reinforcement of taste sensations throughout the mouth, not just on the tongue. The participants evaluated the three attributes of the test soups on a 5-point rating scale ranging from –2 (apparently suppressed) to +2 (apparently strong). Additionally, the palatability of each sample was evaluated on a similar scale ranging from –2 (apparently bad) to +2 (apparently good). All the values for the three kokumi attributes and palatability were set to 0 for the control soup.

### Statistical analyses

Data are presented as the mean ± standard error of the mean (SEM). A Shapiro–Wilk test was performed to confirm that the data were normally distributed. Student's $t$-tests (paired, two-tailed) were used to assess statistical differences between two groups. Additionally, to analyze more than three groups, we used a one-way ANOVA or repeated-measures two-way ANOVA with post hoc Bonferroni tests to account for multiple comparisons. All statistical analyses were conducted in SPSS Statistics Version 25.0 (IBM, Armonk, NY, USA). Statistical significance was set at $p < 0.05$, except when the Bonferroni correction was used.

## Additional information

### Funding

| Funder | Grant reference number | Author |
| --- | --- | --- |
| Japan Society for the Promotion of Science | JP17K00835 | Takashi Yamamoto |
| Japan Society for the Promotion of Science | JP22K11803 | Kayoko Ueji |

| Funder | Grant reference number | Author |
| --- | --- | --- |

The funders had no role in study design, data collection and interpretation, or the decision to submit the work for publication.

## Author contributions

Takashi Yamamoto, Conceptualization, Data curation, Formal analysis, Funding acquisition, Investigation, Writing - original draft, Writing - review and editing; Kayoko Ueji, Chizuko Inui-Yamamoto, Formal analysis; Haruno Mizuta, Investigation; Natsuko Kumamoto, Yasuhiro Shibata, Formal analysis, Investigation; Shinya Ugawa, Data curation, Formal analysis, Investigation, Writing - original draft

## Author ORCIDs

Takashi Yamamoto https://orcid.org/0000-0003-0132-2936

## Ethics

We recruited 22 participants (19 women and three men, aged 21–28 years) from Kio University who were not affiliated with our laboratory, including students and staff members. All participants passed a screening test based on taste sensitivity. According to the responses obtained from a pre-experimental questionnaire, we confirmed that none of the participants had any sensory abnormalities, eating disorders, or mental disorders, or were taking any medications that may potentially affect their sense of taste. We provided them with a detailed explanation of the experimental procedures, including safety measures and personal data protection, without revealing the specific goals of the study. Thereafter, each participant provided written informed consent. This study was approved by the Ethics Committee of Kio University (approval no. H30-10), and all experiments adhered to the principles of the Declaration of Helsinki.

All animal care and experimental procedures were performed in strict accordance with the National Institutes of Health guidelines. Experimental protocols were approved by the Institutional Animal Care and Use Committee of Kio University (protocol no. H28-01). All surgery was performed under anesthesia with a combination anesthetic (medetomidine, midazolam, and butorphanol), and every effort was made to minimize suffering.

Reviewer #1 (Public review): https://doi.org/10.7554/eLife.101629.4.sa1
Reviewer #2 (Public review): https://doi.org/10.7554/eLife.101629.4.sa2
Reviewer #3 (Public review): https://doi.org/10.7554/eLife.101629.4.sa3
Author response https://doi.org/10.7554/eLife.101629.4.sa4

# Additional files

## Supplementary files
MDAR checklist

## Data availability

All data generated or analysed during this study are available within the manuscript and supporting files; source data files have been provided for Figures 1-5 and the figure supplements. These files contain the numerical data used to generate the figures.

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
