## [Editor Report · eLife Assessment]

In this **valuable** study, the authors used rats to determine the receptor for a food-related perception that has been characterized in humans. The data are **solid** in terms of methods and analysis: the data show that this stimulus (ornithine) has some additive effects in terms of increasing preference and taste response in rats when it is mixed with other more common taste stimuli. Therefore, the combinations of experiments generally support (but do not conclusively prove) the hypothesis that the ‘kokumi’ taste effect elicited by this stimulus in humans may be mediated by the specific receptor examined in the study.

---

## [Referee Report · Reviewer #1 (Public review)]

Summary:

This paper contains what could be described as a "classic" approach towards evaluating a novel taste stimuli in an animal model, including standard behavioral tests (some with nerve transections), taste nerve physiology, and immunocytochemistry of taste cells of the tongue. The stimulus being tested is ornithine, from a class of stimuli called "kokumi" (in terms of human taste); these kokumi stimuli appear to enhance other canonical tastes, increasing what are essentially hedonic attributes of other stimuli. The mechanism for ornithine detection is thought to be GPRC6A receptors expressed in taste cells. The authors showed evidence for this in an earlier paper with mice; this paper evaluates ornithine taste in a rat model, and comes to a similar conclusion, albeit with some small differences between the two rodent species.

Strengths:

The data show effects of ornithine on taste/intake in laboratory rats: In two-bottle and briefer intake tests, adding ornithine results in higher intake of most, but all not all stimuli tested. Bilateral chorda tympani (CT) nerve cuts or the addition of GPRC6A antagonists decreased or eliminated these effects. Ornithine also evoked responses by itself in the CT nerve, but mainly at higher concentrations; at lower concentrations it potentiated the response to monosodium glutamate. Finally, immunocytochemistry of taste cell expression indicated that GPRC6A was expressed predominantly in the anterior tongue, and co-localized (to a small extent) with only IP3R3, indicative of expression in a subset of type II taste receptor cells.

Weaknesses:

As the authors are aware, it is difficult to assess a complex human taste with complex attributes, such as kokumi, in an animal model. In these experiments they attempt to uncover mechanistic insights about how ornithine potentiates other stimuli by using a variety of established experimental approaches in rats. They partially succeed by finding evidence that GPRC6A may mediate effects of ornithine when it is used at lower concentrations. In the revisions they have scaled back their interpretations accordingly. A supplementary experiment measuring certain aspects of the effects of ornithine added to Miso soup in human subjects is included for the express purpose of establishing that the kokumi sensation of a complex solution is enhanced by ornithine. This (supplementary) experiment was conducted with a small sample size, and though perhaps useful, these preliminary results do not align particularly well with the animal experiments. It would be helpful to further explore human taste of ornithine in a larger and better-controlled study.

---

## [Referee Report · Reviewer #2 (Public review)]

Summary:

The authors used rats to determine the receptor for a food-related perception (kokumi) that has been characterized in humans. They employ a combination of behavioral, electrophysiological, and immunohistochemical results to support their conclusion that ornithine-mediated kokumi effects are mediated by the GPRC6A receptor. They complemented the rat data with some human psychophysical data. I find the results intriguing, but believe that the authors overinterpret their data.

Strengths:

The authors provide compelling evidence that ornithine enhances the palatability of several chemical stimuli (i.e., IMP, MSG, MPG, Intralipos, sucrose, NaCl, quinine). Ornithine also increases CT nerve responses to MSG. Additionally, the authors provide evidence that the effects of ornithine are mediated by GPRC6A, a G-protein-coupled receptor family C group 6 subtype A, and that this receptor is expressed primarily in fungiform taste buds. Taken together, these results indicate that ornithine enhances the palatability of multiple taste stimuli in rats, and that the enhancement is mediated, at least in part, within fungiform taste buds. This finding could stand on its own. The question of whether ornithine produces these effects by eliciting kokumi-like perceptions (see below) should be presented as speculation in the Discussion section.

Weaknesses:

I am still unconvinced that the measurements in rats reflect the "kokumi" taste percept described in humans. The authors conducted long-term preference tests, 10-min avidity tests and whole chorda tympani (CT) nerve recordings. None of these procedures specifically model features of "kokumi" perception in humans, which (according to the authors) include increasing "intensity of whole complex tastes (rich flavor with complex tastes), mouthfulness (spread of taste and flavor throughout the oral cavity), and persistence of taste (lingering flavor)." While it may be possible to develop behavioral assays in rats (or mice) that effectively model kokumi taste perception in humans, the authors have not made any effort to do so. As a result, I do not think that the rat data provide support for the main conclusion of the study--that "ornithine is a kokumi substance and GPRC6A is a novel kokumi receptor."

Why are the authors hypothesizing that the primary impacts of ornithine are on the peripheral taste system? While the CT recordings provide support for peripheral taste enhancement, they do not rule out the possibility of additional central enhancement. Indeed, based on the definition of human kokumi described above, it is likely that the effects of kokumi stimuli in humans are mediated at least in part by the central flavor system.

The authors include (in the supplemental data section) a pilot study that examined the impact of ornithine on variety of subjective measures of flavor perception in humans. The presence of this pilot study within the larger rat study does not really make sense. If the human studies are so important, as the authors state, then why did the authors relegate them to the supplemental data section? Usually one places background and negative findings in this section of a paper. Accordingly, I recommend that the human data be published in a separate article.

---

## [Referee Report · Reviewer #3 (Public review)]

Summary:

In this study the authors set out to investigate whether GPRC6A mediates kokumi taste initiated by the amino acid L-ornithine. They used Wistar rats, a standard laboratory strain, as the primary model and also performed an informative taste test in humans, in which miso soup was supplemented with various concentrations of L-ornithine. The findings are valuable and overall the evidence is solid. L-Ornithine should be considered to be a useful test substance in future studies of kokumi taste and the class C G protein coupled receptor known as GPRC6A (C6A) along with its homolog, the calcium-sensing receptor (CaSR) should be considered candidate mediators of kokumi taste. The researchers confirmed in rats their previous work on Ornithine and C6A in mice (Mizuta et al Nutrients 2021).

Strengths:

The overall experimental design is solid based on two bottle preference tests in rats. After determining the optimal concentration for L-Ornithine (1 mM) in the presence of MSG, it was added to various tastants including: inosine 5'-monophosphate; monosodium glutamate (MSG); mono-potassium glutamate (MPG); intralipos (a soybean oil emulsion); sucrose; sodium chloride (NaCl; salt); citric acid (sour) and quinine hydrochloride (bitter). Robust effects of ornithine were observed in the cases of IMP, MSG, MPG and sucrose; and little or no effects were observed in the cases of sodium chloride, citric acid; quinine HCl. The researchers then focused on the preference for Ornithine-containing MSG solutions. Inclusion of the C6A inhibitors Calindol (0.3 mM but not 0.06 mM) or the gallate derivative EGCG (0.1 mM but not 0.03 mM) eliminated the preference for solutions that contained Ornithine in addition to MSG. The researchers next performed transections of the chord tympani nerves (with sham operation controls) in anesthetized rats to identify a role of the chorda tympani branches of the facial nerves (cranial nerve VII) in the preference for Ornithine-containing MSG solutions. This finding implicates the anterior half-two thirds of the tongue in ornithine-induced kokumi taste. They then used electrical recordings from intact chorda tympani nerves in anesthetized rats to demonstrate that ornithine enhanced MSG-induced responses following the application of tastants to the anterior surface of the tongue. They went on to show that this enhanced response was insensitive to amiloride, selected to inhibit 'salt tastant' responses mediated by the epithelial Na+ channel, but eliminated by Calindol. Finally they performed immunohistochemistry on sections of rat tongue demonstrating C6A positive spindle-shaped cells in fungiform papillae that partially overlapped in its distribution with the IP3 type-3 receptor, used as a marker of Type-II cells, but not with (i) gustducin, the G protein partner of Tas1 receptors (T1Rs), used as a marker of a subset of type-II cells; or (ii) 5-HT (serotonin) and Synaptosome-associated protein 25 kDa (SNAP-25) used as markers of Type-III cells.

At least two other receptors in addition to C6A might mediate taste responses to ornithine: (i) the CaSR, which binds and responds to multiple L-amino acids (Conigrave et al, PNAS 2000), and which has been previously reported to mediate kokumi taste (Ohsu et al., JBC 2010) as well as responses to Ornithine (Shin et al., Cell Signaling 2020); and (ii) T1R1/T1R3 heterodimers which also respond to L-amino acids and exhibit enhanced responses to IMP (Nelson et al., Nature 2001). These alternatives are appropriately discussed and, taken together, the experimental results favor the authors' interpretation that C6A mediates the Ornithine responses. The authors provide preliminary data in Suppl. 3 for the possibility of co-expression of C6A with the CaSR.

In the Discussion, the authors consider the potential effects of kokumi substances on the threshold concentrations of key tastants such as glutamate, arguing that extension of taste distribution to additional areas of the mouth (previously referred to as 'mouthfulness') and persistence of taste/flavor responses (previously referred to as 'continuity') could arise from a reduction in the threshold concentrations of umami and other substances that evoke taste responses. This concept may help to design future experiments.

Weaknesses:

The authors point out that animal models pose some difficulties of interpretation in studies of taste and raise the possibility in the Discussion that umami substances may enhance the taste response to ornithine (Line 271, Page 9).

The status of one of the compounds used as an inhibitor of C6A, the gallate derivative EGCG, as a potential inhibitor of the CaSR or T1R1/T1R3 is unknown. It would have been helpful to show that a specific inhibitor of the CaSR failed to block the ornithine response.

It would have been helpful to include a positive control kokumi substance in the two bottle preference experiment (e.g., one of the known gamma glutamyl peptides such as gamma-glu-Val-Gly or glutathione), to compare the relative potencies of the control kokumi compound and Ornithine, and to compare the sensitivities of the two responses to C6A and CaSR inhibitors.

---

## [Author Response]

The following is the authors’ response to the previous reviews

**Public Reviews:**

**Reviewer #1 (Public review):**
Summary:This paper contains what could be described as a "classic" approach towards evaluating a novel taste stimuli in an animal model, including standard behavioral tests (some with nerve transections), taste nerve physiology, and immunocytochemistry of taste cells of the tongue. The stimulus being tested is ornithine, from a class of stimuli called "kokumi" (in terms of human taste); these kokumi stimuli appear to enhance other canonical tastes, increasing what are essentially hedonic attributes of other stimuli. The mechanism for ornithine detection is thought to be GPRC6A receptors expressed in taste cells. The authors showed evidence for this in an earlier paper with mice; this paper evaluates ornithine taste in a rat model, and comes to a similar conclusion, albeit with some small differences between the two rodent species.Strengths:The data show effects of ornithine on taste/intake in laboratory rats: In two-bottle and briefer intake tests, adding ornithine results in higher intake of most, but all not all stimuli tested. Bilateral chorda tympani (CT) nerve cuts or the addition of GPRC6A antagonists decreased or eliminated these effects. Ornithine also evoked responses by itself in the CT nerve, but mainly at higher concentrations; at lower concentrations it potentiated the response to monosodium glutamate. Finally, immunocytochemistry of taste cell expression indicated that GPRC6A was expressed predominantly in the anterior tongue, and co-localized (to a small extent) with only IP3R3, indicative of expression in a subset of type II taste receptor cells.Weaknesses:As the authors are aware, it is difficult to assess a complex human taste with complex attributes, such as kokumi, in an animal model. In these experiments they attempt to uncover mechanistic insights about how ornithine potentiates other stimuli by using a variety of established experimental approaches in rats. They partially succeed by finding evidence that GPRC6A may mediate effects of ornithine when it is used at lower concentrations. In the revision they have scaled back their interpretations accordingly. A supplementary experiment measuring certain aspects of the effects of ornithine added to Miso soup in human subjects is included for the express purpose of establishing that the kokumi sensation of a complex solution is enhanced by ornithine; however, they do not use any such complex solutions in the rat studies. Moreover, the sample size of the human experiment is (still) small - it really doesn't belong in the same manuscript with the rat studies.

Despite the reviewer’s suggestion, we would like to include the human sensory experiment. Our rationale is that we must first demonstrate that the kokumi of miso soup is enhanced by the addition of ornithine, which is then followed by basic animal experiments to investigate the underlying mechanisms of kokumi in humans.

We did not present the additive effects of ornithine on miso soup in the present rat study because our previous companion paper (Fig. 1B in Mizuta et al., 2021, Ref. #26) already confirmed that miso soup supplemented with 3 mM L-ornithine (but not D-ornithine) was statistically significantly (P < 0.001) preferred to plain miso soup by mice.

Furthermore, we believe that our sample size (n = 22) is comparable to those employed in other studies. For example, the representative kokumi studies by Ohsu et al. (Ref. #9), Ueda et al. (Ref. #10), Shibata et al. (Ref. #20), Dunkel et al. (Ref. #37), and Yang et al. (Ref. #44) used sample sizes of 20, 19, 17, 9, and 15, respectively.

**Reviewer #2 (Public review):**
Summary:The authors used rats to determine the receptor for a food-related perception (kokumi) that has been characterized in humans. They employ a combination of behavioral, electrophysiological, and immunohistochemical results to support their conclusion that ornithine-mediated kokumi effects are mediated by the GPRC6A receptor. They complemented the rat data with some human psychophysical data. I find the results intriguing, but believe that the authors overinterpret their data.Strengths:The authors provide compelling evidence that ornithine enhances the palatability of several chemical stimuli (i.e., IMP, MSG, MPG, Intralipos, sucrose, NaCl, quinine). Ornithine also increases CT nerve responses to MSG. Additionally, the authors provide evidence that the effects of ornithine are mediated by GPRC6A, a G-protein-coupled receptor family C group 6 subtype A, and that this receptor is expressed primarily in fungiform taste buds. Taken together, these results indicate that ornithine enhances the palatability of multiple taste stimuli in rats and that the enhancement is mediated, at least in part, within fungiform taste buds. This is an important finding that could stand on its own. The question of whether ornithine produces these effects by eliciting kokumi-like perceptions (see below) should be presented as speculation in the Discussion section.Weaknesses:I am still unconvinced that the measurements in rats reflect the "kokumi" taste percept described in humans. The authors conducted long-term preference tests, 10-min avidity tests and whole chorda tympani (CT) nerve recordings. None of these procedures specifically model features of "kokumi" perception in humans, which (according to the authors) include increasing "intensity of whole complex tastes (rich flavor with complex tastes), mouthfulness (spread of taste and flavor throughout the oral cavity), and persistence of taste (lingering flavor)." While it may be possible to develop behavioral assays in rats (or mice) that effectively model kokumi taste perception in humans, the authors have not made any effort to do so. As a result, I do not think that the rat data provide support for the main conclusion of the study--that "ornithine is a kokumi substance and GPRC6A is a novel kokumi receptor."

Kokumi can be assessed in humans, as demonstrated by the enhanced kokumi perception observed when miso soup is supplemented with ornithine (Fig. S1). Currently, we do not have a method to measure the same kokumi perception in animals. However, in the two-bottle preference test, our previous companion paper (Fig. 1B in Mizuta et al. 2021, Ref. #26) confirmed that miso soup supplemented with 3 mM L-ornithine (but not D-ornithine) was statistically significantly (P < 0.001) preferred over plain miso soup by mice.

Of the three attributes of kokumi perception in humans, the “intensity of whole complex tastes (rich flavor with complex tastes)” was partly demonstrated in the present rat study. In contrast, “mouthfulness (the spread of taste and flavor throughout the oral cavity)” could not be directly detected in animals and had to be inferred in the Discussion. “Persistence of taste (lingering flavor)” was evident at least in the chorda tympani responses; however, because the tongue was rinsed 30 seconds after the onset of stimulation, the duration of the response was not fully recorded.

It is well accepted in sensory physiology that the stronger the stimulus, the larger the tonic response—and consequently, the longer it takes for the response to return to baseline. For example, Kawasaki et al. (2016, Ref. #45) clearly showed that the duration of sensation increased proportionally with the concentration of MSG, lactic acid, and NaCl in human sensory tests. The essence of this explanation has been incorporated into the Discussion (p. 12).

Why are the authors hypothesizing that the primary impacts of ornithine are on the peripheral taste system? While the CT recordings provide support for peripheral taste enhancement, they do not rule out the possibility of additional central enhancement. Indeed, based on the definition of human kokumi described above, it is likely that the effects of kokumi stimuli in humans are mediated at least in part by the central flavor system.

We agree with the reviewer’s comment. Our CT recordings indicate that the effects of kokumi stimuli on taste enhancement occur primarily at the peripheral taste organs. The resulting sensory signals are then transmitted to the brain, where they are processed by the central gustatory and flavor systems, ultimately giving rise to kokumi attributes. This central involvement in kokumi perception is discussed on page 12. Although kokumi substances exert their effects at low concentrations—levels at which the substance itself (e.g., ornithine) does not become more favorable or (in the case of γ-Glu-Val-Gly) exhibits no distinct taste—we cannot rule out the possibility that even faint taste signals from these substances are transmitted to the brain and interact with other taste modalities.

The authors include (in the supplemental data section) a pilot study that examined the impact of ornithine on variety of subjective measures of flavor perception in humans. The presence of this pilot study within the larger rat study does not really mice sense. While I agree with the authors that there is value in conducting parallel tests in both humans and rodents, I think that this can only be done effectively when the measurements in both species are the same. For this reason, I recommend that the human data be published in a separate article.

Despite the reviewer’s suggestion, we intend to include the human sensory experiment. Our rationale is that we must first demonstrate that the kokumi of miso soup is enhanced by the addition of ornithine, and then follow up with basic animal experiments to investigate the potential underlying mechanisms of kokumi in humans.

In our previous companion paper (Fig. 1B in Mizuta et al., 2021, Ref. #26), we confirmed with statistical significance (P < 0.001) that mice preferred miso soup supplemented with 3 mM L-ornithine (but not D-ornithine) over plain miso soup. However, as explained in our response to Reviewer #2’s first concern (in the Public review), it is difficult to measure two of the three kokumi attributes—aside from the “intensity of whole complex tastes (rich flavor with complex tastes)”—in animal models.

The authors indicated on several occasions (e.g., see Abstract) that ornithine produced "synergistic" effects on the CT nerve response to chemical stimuli. "Synergy" is used to describe a situation where two stimuli produce an effect that is greater than the sum of the response to each stimulus alone (i.e., 2 + 2 = 5). As far as I can tell, the CT recordings in Fig. 3 do not reflect a synergism.

We appreciate your comments regarding the definition of synergy. In Fig. 5 (not Fig. 3), please note the difference in the scaling of the ordinate between Fig. 5D (ornithine responses) and Fig. 5E (MSG responses). When both responses are presented on the same scale, it becomes evident that the response to 1 mM ornithine is negligibly small compared to the MSG response, which clearly indicates that the response to the mixture of MSG and 1 mM ornithine exceeds the sum of the individual responses to MSG and 1 mM ornithine. Therefore, we have described the effect as “synergistic” rather than “additive.” The same observation applies to the mice experiments in our previous companion paper (Fig. 8 in Mizuta et al. 2021, Ref. #26), where synergistic effects are similarly demonstrated by graphical representation. We have also added the following sentence to the legend of Fig. 5:

“Note the different scaling of the ordinate in (D) and (E).”

**Reviewer #3 (Public review):**
Summary:In this study the authors set out to investigate whether GPRC6A mediates kokumi taste initiated by the amino acid L-ornithine. They used Wistar rats, a standard laboratory strain, as the primary model and also performed an informative taste test in humans, in which miso soup was supplemented with various concentrations of L-ornithine. The findings are valuable and overall the evidence is solid. L-Ornithine should be considered to be a useful test substance in future studies of kokumi taste and the class C G protein coupled receptor known as GPRC6A (C6A) along with its homolog, the calcium-sensing receptor (CaSR) should be considered candidate mediators of kokumi taste. The researchers confirmed in rats their previous work on Ornithine and C6A in mice (Mizuta et al Nutrients 2021).Strengths:The overall experimental design is solid based on two bottle preference tests in rats. After determining the optimal concentration for L-Ornithine (1 mM) in the presence of MSG, it was added to various tastants including: inosine 5'-monophosphate; monosodium glutamate (MSG); mono-potassium glutamate (MPG); intralipos (a soybean oil emulsion); sucrose; sodium chloride (NaCl; salt); citric acid (sour) and quinine hydrochloride (bitter). Robust effects of ornithine were observed in the cases of IMP, MSG, MPG and sucrose; and little or no effects were observed in the cases of sodium chloride, citric acid; quinine HCl. The researchers then focused on the preference for Ornithine-containing MSG solutions. Inclusion of the C6A inhibitors Calindol (0.3 mM but not 0.06 mM) or the gallate derivative EGCG (0.1 mM but not 0.03 mM) eliminated the preference for solutions that contained Ornithine in addition to MSG. The researchers next performed transections of the chord tympani nerves (with sham operation controls) in anesthetized rats to identify a role of the chorda tympani branches of the facial nerves (cranial nerve VII) in the preference for Ornithine-containing MSG solutions. This finding implicates the anterior half-two thirds of the tongue in ornithine-induced kokumi taste. They then used electrical recordings from intact chorda tympani nerves in anesthetized rats to demonstrate that ornithine enhanced MSG-induced responses following the application of tastants to the anterior surface of the tongue. They went on to show that this enhanced response was insensitive to amiloride, selected to inhibit 'salt tastant' responses mediated by the epithelial Na+ channel, but eliminated by Calindol. Finally they performed immunohistochemistry on sections of rat tongue demonstrating C6A positive spindle-shaped cells in fungiform papillae that partially overlapped in its distribution with the IP3 type-3 receptor, used as a marker of Type-II cells, but not with (i) gustducin, the G protein partner of Tas1 receptors (T1Rs), used as a marker of a subset of type-II cells; or (ii) 5-HT (serotonin) and Synaptosome-associated protein 25 kDa (SNAP-25) used as markers of Type-III cells.At least two other receptors in addition to C6A might mediate taste responses to ornithine: (i) the CaSR, which binds and responds to multiple L-amino acids (Conigrave et al, PNAS 2000), and which has been previously reported to mediate kokumi taste (Ohsu et al., JBC 2010) as well as responses to Ornithine (Shin et al., Cell Signaling 2020); and (ii) T1R1/T1R3 heterodimers which also respond to L-amino acids and exhibit enhanced responses to IMP (Nelson et al., Nature 2001). These alternatives are appropriately discussed and, taken together, the experimental results favor the authors' interpretation that C6A mediates the Ornithine responses. The authors provide preliminary data in Suppl. 3 for the possibility of co-expression of C6A with the CaSR.Weaknesses:The authors point out that animal models pose some difficulties of interpretation in studies of taste and raise the possibility in the Discussion that umami substances may enhance the taste response to ornithine (Line 271, Page 9).

Ornithine and umami substances interact to produce synergistic effects in both directions—ornithine enhances responses to umami substances, and vice versa. These effects may depend on the concentrations used, as described in the Discussion (pp. 9–10). Further studies are required to clarify the precise nature of this interaction.

One issue that is not addressed, and could be usefully addressed in the Discussion, relates to the potential effects of kokumi substances on the threshold concentrations of key tastants such as glutamate. Thus, an extension of taste distribution to additional areas of the mouth (previously referred to as 'mouthfulness') and persistence of taste/flavor responses (previously referred to as 'continuity') could arise from a reduction in the threshold concentrations of umami and other substances that evoke taste responses.

Thank you for this important suggestion. If ornithine reduces the threshold concentrations of tastants—including glutamate—and enhances their suprathreshold responses, then adding ornithine may activate additional taste cells. This effect could explain kokumi attributes such as an “extension of taste distribution” and possibly the “persistence of responses.” As shown in Fig. 2, the lowest concentrations used for each taste stimulus are near or below the thresholds, which indicates that threshold concentrations are reduced—especially for MSG and MPG. We have incorporated this possibility into the Discussion as follows (p.12):

“Kokumi substances may reduce the threshold concentrations as well as they increase the suprathreshold responses of tastants. Once the threshold concentrations are lowered, additional taste cells in the oral cavity become activated, and this information is transmitted to the brain. As a result, the brain perceives this input as coming from a wider area of the mouth.”

The status of one of the compounds used as an inhibitor of C6A, the gallate derivative EGCG, as a potential inhibitor of the CaSR or T1R1/T1R3 is unknown. It would have been helpful to show that a specific inhibitor of the CaSR failed to block the ornithine response.

Thank you for this important comment. We attempted to identify a specific inhibitor of CaSR. Although we considered using NPS-2143—a commonly used CaSR inhibitor—it is known to also inhibit GPRC6A. We agree that using a specific CaSR inhibitor would be beneficial and plan to pursue this in future studies.

It would have been helpful to include a positive control kokumi substance in the two bottle preference experiment (e.g., one of the known gamma glutamyl peptides such as gamma-glu-Val-Gly or glutathione), to compare the relative potencies of the control kokumi compound and Ornithine, and to compare the sensitivities of the two responses to C6A and CaSR inhibitors.

We agree with this comment. In retrospect, it may have been advantageous to directly compare the potencies of CaSR and GPRC6A agonists in enhancing taste preferences—and to evaluate the sensitivity of these preferences to CaSR and GPRC6A antagonists. However, we did not include γ-Glu-Val-Gly in the present study because we have already reported its supplementation effects on the ingestion of basic taste solutions in rats using the same methodology in a separate paper (Yamamoto and Mizuta, 2022, Ref. #25). The results from both studies are compared in the Discussion (p. 11).

**Recommendations for the authors:**

**Reviewer #1 (Recommendations for the authors):**
Major:I am not convinced by the Author's arguments for including the human data. I appreciate their efforts in adding a few (5) subjects and improving the description, but it still feels like it is shoehorned into this paper, and would be better published as a different manuscript.

This human study is short, but it is complete rather than preliminary. The rationale for us to include the human data as supplementary information is shown in responses to the reviewer’s Public review.

Minor concerns:Page 3 paragraph 1: Suggest "contributing to palatability".

Thank you for this suggestion. We have rewritten the text as follows:

“…, the brain further processes these sensations to evoke emotional responses, contributing to palatability or unpleasantness”.

Page 4 paragraph 2: The text still assumes that "kokumi" is a meaningful descriptor for what rodents experience. Re-wording the following sentence like this could help:"Neuroscientific studies in mice and rats provide evidence that gluthione and y-Glu-Val-Gly activate CaSRs, and modify behavioral responses to other tastants in a way that may correspond to kokumi taste as experienced by humans. However, to our..."Or something similar.

Thank you for this suggestion. We have rewritten the sentence according to your suggestion as follows:

"Neuroscientific studies (23,25,30) in mice and rats provide evidence that glutathione and y-Glu-Val-Gly activate CaSRs, and modify behavioral responses to other tastants in a way that may correspond to kokumi as experienced by humans”.

Page 7 paragraph 1 - put the concentrations of Calindol and EGCG used (in the physiology exps) in the text.

We have added the concentrations: “300 µM calindol and 100 µM EGCG”.

**Reviewer #2 (Recommendations for the authors):**
I have included all of my recommendations in the public review section.
**Reviewer #3 (Recommendations for the authors):**
Although the definitions of 'thickness', 'mouthfulness' and 'continuity' have been revised very helpfully in the Introduction, 'mouthfulness' reappears at other points in the MS e.g., Page 4, Results, Line 3; Page 9, Line 3. It is best replaced by the new definition in these other locations too.

We wish to clarify that our revised text stated, “…to clarify that kokumi attributes are inherently gustatory, in the present study we use the terms ‘intensity of whole complex tastes (rich flavor with complex tastes)’ instead of ‘thickness,’ ‘mouthfulness (spread of taste and flavor throughout the oral cavity)’ instead of ‘continuity,’ and ‘persistence of taste (lingering flavor)’ instead of ‘continuity.’” The term “mouthfulness” was retained in our text, though we provided a more specific explanation. In the re-revised version, we have added “(spread of taste in the oral cavity)” immediately after “mouthfulness.”

I doubt that many scientific readers will be familliar with the term 'intragemmal nerve fibres' (Page 8, Line 4). It is used appropriately but it would be helpful to briefly define/explain it.

We have added an explanation as follows:

“… intragemmal nerve fibers, which are nerve processes that extend directly into the structure of the taste bud to transmit taste signals from taste cells to the brain.”

I previously pointed out the overlap between the CaSR's amino acid (AA) and gamma-glutamyl-peptide binding site. I was surprised by the authors' response which appeared to miss the point being made. It was based on the impacts of selected mutations in the receptor's Venus FlyTrap domain (Broadhead JBC 2011) on the responses to AAs and glutathione analogs. The significantly more active analog, S-methylglutathione is of additional interest because, like glutathione itself, it is present in mammalian body fluids. My apologies to the authors for not more carefully explaining this point.

Thank you for this comment. Both CaSR and GPRC6A are recognized as broad-spectrum amino acid sensors; however, their agonist profiles differ. Aromatic amino acids preferentially activate CaSR, whereas basic amino acids tend to activate GPRC6A. For instance, among basic amino acids, ornithine is a potent and specific activator of GPRC6A, while γ-Glu-Val-Gly in addition to amino acids is a high-potency activator of CaSR. It remains unclear how effectively ornithine activates CaSR and whether γ-glutamyl peptides also activate GPRC6A. These questions should be addressed in future studies.